# Seasonal Changes in Serum Metabolites in Multiple Sclerosis Relapse

**DOI:** 10.3390/ijms24043542

**Published:** 2023-02-10

**Authors:** Ekaterina Martynova, Timur Khaibullin, Ilnur Salafutdinov, Maria Markelova, Alexander Laikov, Leonid Lopukhov, Rongzeng Liu, Kritika Sahay, Mehendi Goyal, Manoj Baranwal, Albert A Rizvanov, Svetlana Khaiboullina

**Affiliations:** 1Institute of Fundamental Medicine and Biology, Kazan (Volga Region) Federal University, 420008 Kazan, Russia; 2Republican Clinical Neurological Center, Republic of Tatarstan, 420021 Kazan, Russia; 3Department of Medical Biology and Genetic, Kazan State Medical University, 420088 Kazan, Russia; 4Department of Immunology, School of Basic Medical Sciences, Henan University of Science and Technology, Luoyang 471003, China; 5Department of Biotechnology, Thapar Institute of Engineering and Technology, Patiala 147004, India

**Keywords:** multiple sclerosis, serum, metabolites, autoimmune diseases

## Abstract

Multiple sclerosis (MS) is a debilitating chronic disease of unknown etiology. There are limited treatment options due to an incomplete understanding of disease pathology. The disease is shown to have seasonal exacerbation of clinical symptoms. The mechanisms of such seasonal worsening of symptoms remains unknown. In this study, we applied targeted metabolomics analysis of serum samples using LC-MC/MC to determine seasonal changes in metabolites throughout the four seasons. We also analyzed seasonal serum cytokine alterations in patients with relapsed MS. For the first time, we can demonstrate seasonal changes in various metabolites in MS compared to the control. More metabolites were affected in MS in the fall season followed by spring, while summer MS was characterized by the smallest number of affected metabolites. Ceramides were activated in all seasons, suggesting their central role in the disease pathogenesis. Substantial changes in glucose metabolite levels were found in MS, indicating a potential shift to glycolysis. An increased serum level of quinolinic acid was demonstrated in winter MS. Histidine pathways were affected, suggesting their role in relapse of MS in the spring and fall. We also found that spring and fall seasons had a higher number of overlapping metabolites affected in MS. This could be explained by patients having a relapse of symptoms during these two seasons.

## 1. Introduction

Multiple sclerosis (MS) is a disease that introduces many challenges for healthcare providers. The disease presents in multiple forms, making diagnosis and disease progression predictions difficult [1]. These forms include four categories of clinical manifestation: relapsing–remitting MS (RRMS), secondary progressive MS (SPMS), primary progressive MS (PPMS), and progressive relapsing MS (PRMS) [2]. The most common form is RRMS, which is diagnosed in 85% of all MS patients. This form is characterized by exacerbations followed by periods of complete remission. SPMS starts as PRMS with symptoms gradually worsening without periods of remission. PPMS is characterized by progressive deterioration of symptoms without remission. PRMS is a rare form in which the disease has progressive development from disease onset with continuously worsening symptoms. MS further complicates treatment due to its resistance to therapy, leaving limited options for disease management. 

From the point of diagnosis, the course of MS can vary significantly among patients, making early prediction of the clinical form of disease challenging [2]. This affects the selection of disease modifying therapies. Factors contributing to disease severity include the sex and age of the patient, in which older males have a higher chance of poor prognosis [3,4]. Another factor, vitamin D (vitD), has been shown to have a prognostic value in the early stages of the disease, in which lower levels were linked to a higher risk of relapse [5]. A study of the MS metabolome has demonstrated phosphatidylcholine proteins as potential biomarkers for PPMS diagnosis and prognostic markers for disease progression [6]. In another study, aberrant serum levels of kynurenic and quinolinic acids, metabolites of the kynurenine pathway, were shown to have a prognostic value in MS, specifically in determining the switch to progressive forms of the disease [7]. Despite these study findings, using identified biomarkers to diagnose and predict disease progression remains limited as there are still major gaps in understanding disease pathogenesis. 

There are multiple factors that have been found to exacerbate MS symptoms, with the spring and fall seasons associated with a higher risk of relapse and/or T2 activity in the brain [8]. This is supported by a seasonal biphasic pattern of MS relapses in an Italian cohort of patients [9]. Harding et al. have demonstrated similar seasonal patterns of MS relapse in the United Kingdom [10]. However, Ma and Zhang have demonstrated a higher frequency of MS symptoms exacerbation in the winter as compared to the summer [11]. In a Brazilian cohort, the relapses and exacerbation of symptoms did not follow the same seasonal pattern, suggesting additional environmental factors contributing to MS activity [12]. A lack of seasonal occurrence of relapses in several studies also support the multifactorial role of MS pathogenesis [13,14,15]. The contribution of seasons to the course of MS is a consistent feature of the disease, and it has been suggested to include seasonality as part of clinical trial design [16]. Although seasonal exacerbations are well documented, there is little known about biomarkers that could predict or explain them. 

Seasonal changes in MS symptoms reflect alterations in the serum level of disease markers. This is supported by the discovery of season-dependent changes in serum levels of melatonin, a central hormone and regulator of circadian cycles that is linked to pathogenesis of MS [17,18]. The involvement of another hormone, vitD, in seasonal exacerbations has been shown by Hartl et al. [19]. The underlying disease pathogenesis remains poorly understood and is believed to be multifactorial. Various factors, some of which are metabolites, could contribute to relapses. As expected, metabolites with potential contribution to MS relapse have been previously identified [20,21]. Interestingly, these studies demonstrated significant variations in MS serum metabolites, suggesting that other factors, not only the form of the disease, could contribute to metabolic activity. Some metabolites identified in MS could cause inflammation and activate the immune response [7]; however, their activation in relation to the season remains largely unknown.

Our study aims to analyze the seasonal variation of serum metabolite levels in MS compared to the control. Furthermore, we wanted to identify the season when the largest number of metabolites are affected in MS. In addition, we aimed to demonstrate individual metabolites most affected in MS in different seasons. Finally, we sought to identify metabolites that were most affected in MS during exacerbation.

## 2. Results

### 2.1. Metabolites Analyses

There were 349 metabolites analyzed in serum of MS and controls. Serum levels of 91 metabolites were affected in MS as compared to the control in all seasons (Table 1). Some metabolites were affected in all seasons, while others were affected during specific seasons. These differences in metabolite level were also demonstrated using PLS-DA analysis (Figure 1). We have found seasonal differences in serum levels of metabolites in the control. It appears that metabolites in the winter, spring, and summer cluster closely, while in fall they form a distant group (Figure 1; yellow, light brown, brown, and dark brown). Metabolite levels in MS serum also change depending on the season; the winter, spring, and summer samples form closely located groups, while fall samples show a more dispersed pattern (Figure 1; light blue, blue, dark blue, and darkest blue).

Serum metabolite differences are also evident when each season was analyzed separately (Figure 1A). MS and control metabolites formed separate groups in the spring, with no occurrence of overlapping (Figure 1B). In the summer, metabolites were placed closer with some overlap between the control and MS (Figure 1C). Similarly, in the fall and winter, some metabolites overlapped in MS and the control (Figure 1D,E). Still, in MS and the control, many metabolites were found forming separate groups in the summer, fall, and winter.

#### Hierarchical Cluster Analysis

Hierarchical cluster analysis revealed that metabolites in the control and MS differ, forming two separate branches (Figure 2). This supports our observation that there are substantial differences in serum metabolite levels between MS and the control. It appears the changes in serum metabolites could be clustered into four groups. Group 1 was formed by nine metabolites, which were consistently higher in MS in all four seasons as compared to the control. In Group 2, increased serum levels of MS metabolites were found in the fall, while they were less pronounced in the control. In contrast, Group 3 had an increased serum metabolite levels in the control in the fall as compared to those in MS. Group 4 was formed by metabolites commonly increased in the control but decreased in MS throughout all seasons. 

Interestingly, there were more similarities in affected metabolites in the spring and fall in MS (Figure 2). This is supported by data presented in Figure 3, indicating that 10 metabolites were similarly affected in the spring and fall MS, which is the highest number of similarities between metabolites among all seasons. In contrast, summer and winter metabolites were more similar in the control.

### 2.2. Analysis of Metabolite Levels in MS Based on the Season

We have found seasonal changes in serum levels of 91 metabolites in MS compared to the control. We also found that some metabolites were consistently increased or decreased in MS serum in all seasons (Table 1, Figure 3 “Venn diagrams”). Among these metabolites, sphingosine was increased while three (gentisic acid, DHC (18:1/24:0) and hexacosanoic acid) were decreased in all seasons. There were also metabolites uniquely affected in each season.

***Spring***. We have found that levels of seven metabolites increased in MS in the spring as compared to the control (Appendix A). These metabolites were also found to be increased in some seasons. Three out of seven (42.8%) metabolites were ceramides. Twenty-one metabolites were decreased in MS in the spring as compared to the control. This is the second largest, after the fall season, number of metabolites decreased in MS serum. Most of these metabolites were also decreased in other seasons, one of which was VitD3. Three metabolites were uniquely lowered in the fall only.

***Summer***. Levels of eight metabolites were increased in MS as compared to the control (Appendix A). Two metabolites were uniquely increased in the summer samples, while six were affected in other seasons. Out of eight metabolites, five (62.5%; sphingosine, deoxyguanosine, ceramide (d18:1/24:0), ceramide (d18:1/18:1 OH) and cytidine monophosphate) were also increased in the spring. Additionally, ten metabolites decreased in MS as compared to the control (Figure 2). Two of these were found to be decreased in the summer sample only. Interestingly, six metabolites (gentisic acid, gamma aminobutyric acid, DHC (18:1/24:0), hexacosanoic acid, docosahexaenoic acid and VitD3) were lower in the spring and summer in MS as compared to the control. 

***Fall***. We have found six metabolites increased in MS as compared to the control (Appendix A). One metabolite (niacinamide) was higher in the fall MS only as compared to the control, while five others were affected in other seasons. Two metabolites (sphingosine and ceramide (d18:1/18:1 OH)) were found to be upregulated in the spring, summer, and fall MS. We found fifty-nine metabolites decreased in the fall MS as compared to the related control (Table 1). This is the largest number of metabolites found to be decreased in MS serum among all groups. Interestingly, levels of VitD3 in the fall MS did not differ from that in the control. 

***Winter***. Levels of eighteen metabolites were increased in MS as compared to the control (Appendix A). Most of these metabolites, 12 out of 18 (66.7%), were uniquely affected during this season. The number of metabolites increased in winter MS serum was higher than that in the spring, summer, and fall (seven, eight, and six, respectively). Among the metabolites increased in the winter, levels of six were also increased in the spring, summer, and fall MS as compared to the control. Among these were several ceramides and ceramide (d18:1/18:1). Sphingosine and deoxyguanosine were increased in the spring, fall, and winter MS as compared to the seasonal control.

Serum levels of twelve metabolites were decreased in spring MS as compared to related controls. Twelve metabolites were also decreased in other seasons, one of which was VitD3. One metabolite, 5Hydroxy L-tryptophan, was affected only in the winter.

### 2.3. Metabolic Pathways Affected in MS

Next, metabolites were grouped according to their contribution to various pathways. Overall, we found that more pathways are involved in the fall (eight pathways) (Figure 4). Spring was the second highest season based on the total number of affected pathways (four pathways). This is an interesting observation since this is when the Earth is passing vernal and autumnal equinox. These periods are characterized by rapid changes in the length of the day. These results provide evidence supporting previous observations related to sunlight exposure as a risk factor for MS [22,23,24].

Summer and winter were the least active seasons with only single pathways affected in each season. There are two pathways identified, affected in fall and spring (methylhistidine and beta-alanine) in MS compered to control. The purine pathway was lower in MS compared to control in the spring and summer.

The metabolic network analysis revealed substantial seasonal differences in MS metabolites (Figure 5, Figure 6, Figure 7 and Figure 8). In the spring, two groups of interconnected metabolites could be identified. One group contained two pathways (phosphatidylethanolamine biosynthesis and phosphatidylcholine biosynthesis). There were six pathways in the second group (beta-alanine metabolism, histidine metabolism, ammonia recycling, methionine metabolism, glutamate metabolism, and purine metabolism). Two groups of interconnected metabolites were found in the summer. One group, containing two metabolites, was similar to that in the spring, containing phosphatidylethanolamine biosynthesis and phosphatidylcholine biosynthesis. The second group contained six metabolites (gluconeogenesis, pyruvate metabolism, urea cycle, glutamate metabolism, aspartate metabolism, and purine metabolism). Two pathways (glutamate metabolism and purine metabolism) were similar to that found in spring.

There were more metabolic pathways (28 pathways) forming an interconnected network in the fall. Four of these metabolic pathways were also identified as affected in the spring (beta-alanine metabolism, histidine metabolism, ammonia recycling, and methionine metabolism). The urea cycle and gluconeogenesis pathways were affected in the fall similarly to that in summer MS. A lesser number of pathways (13 pathways) was found to be affected in the winter as compared to fall MS. Six pathways were identified as changed in fall MS as well.

We also analyzed pathways including metabolites shared by spring and fall MS (Figure 9 and Appendix A). Three pathways, such as histidine, beta-alanine, and methyl histidine metabolism, were significantly affected in spring and fall MS. Two pathways, histidine and beta-alanine metabolism, were connected. The methyl histidine metabolism pathway was the most affected in MS. These were seasons when MS patients were diagnosed with the relapse of symptoms. Therefore, it could be suggested that these pathways contribute to pathogenesis of MS exacerbation.

## 3. Discussion

We have identified multiple metabolites that are differentially activated in MS during the four seasons. These differences appear to be significant and could potentially contribute to MS pathogenesis. Our data support research by Zahoor et al. [25] confirming the involvement of the citrate cycle, sphingolipid, and pyruvate metabolism in RRMS. We further added to our understanding of metabolites in pathogenesis of MS. We have demonstrated that serum metabolite levels change depending on the season. The largest number of metabolites affected was found to be in the fall followed by the spring. The smallest number of metabolites changed occurred in the summer.

One of the most exciting observations was identifying several pathways associated with MS exacerbation: histidine, beta-alanine, and methyl histidine metabolism. Among these, histidine and its metabolite, methyl histidine, appear to be the most significantly affected. Our data confirm previous observations of low levels of histidine in serum of MS compared to the control [26,27]. Histidine is a precursor of histamine, a powerful neurotransmitter and immune-modulator [28,29], which was suggested to play a role in MS pathogenesis [30,31]. The immune-modulating role of histidine in pathogenesis of MS could be explained by the inverse correlation between the serum level of this amino acid and IL-6, a pro-inflammatory cytokine [32]. Similarly, low histidine levels were associated with increased serum C-reactive proteins and oxidative stress markers [33]. The role of histidine in pathogenesis of MS was also corroborated using experimental autoimmune encephalitis (EAE), a model of MS [34,35]. Histidine has shown to have neuro-protective properties by reducing the glial scar area and facilitating astrocyte migration to the injury site [36]. Our data suggest that low levels of histidine could reduce the anti-inflammatory effect of this amino acid and its metabolites during the spring and fall, which could contribute to MS exacerbations. The potentials of histidine in MS are currently under investigation in two clinical trials: NCT03266965 and NCT04764383 [37,38]. Our understanding of the seasonal variations of histamine serum level in MS is limited. Studies have identified histidine derivative, trans-urocanic acid, as a photoreceptor in the stratum corneum of skin [39,40]. Ultraviolet light exposure produces cis-urocanic acid, a soluble molecule, which could stimulate oxidative radical production and DNA damage [41,42]. It should be noted that histidine levels change in MS during fall and spring, seasons when sunlight exposure changes substantially. However, the precise mechanisms of these changes in histidine level in MS remain unknown.

Our data confirms previous findings of an increased level of ceramides in MS [43]. Moreover, our data supports Kurz’s et al. suggestion of the role of ceramides in MS pathogenesis [44]. One consistent change in MS metabolites is an increased serum level of ceramides. These data support the previous reports of increased ceramides in MS [44,45] and cerebrospinal fluid (CSF) [43]. Interestingly, we found seasonal variations in ceramide species level in MS. For example, the level of DHC (18:1/24:0) was lower in MS in all seasons. In contrast, ceramide(d18:1/16:0), ceramide (d18:1/18:1 OH), ceramide (d18:1/24:0), ceramide (d18:1/18:1), and ceramide (d18:1/22:0 OH) were increased in selected seasons (Appendix A). These seasonal variations in ceramide levels could explain the conflicting data [45,46]. Ceramides are lipid molecules containing a sphingoid long-chain base linked to an acyl chain with an amide bond [47]. In neurons, ceramides accumulate on the soma and axon, contributing to neuronal adhesion, modulation of ion channels, and neurotransmitter receptors expression [48]. However, an increased level of ceramides could lead to neuronal apoptosis and death due to oxidative stress [49]. This oxidative stress could be induced by virus proteins [49] and pro-inflammatory cytokines [50]. 

Ceramides and sphingosine, increased in MS serum, are intermediate products of the sphingomyelin–ceramide–sphingosine–sphingosine-1 phosphate (S1P) pathway [51]. Our data support the observation made by Wheeler et al., that this pathway appears to be hyperactive in MS [52]. Each step in this pathway is reversible. Ceramides could be produced by hydrolysis of sphingolipids [53]. Ceramide could be metabolized to sphingosine by ceramide synthase [54]. Sphingosine kinase could metabolize sphingosine to S1P [55]. This reaction could be reversed by sphingosine 1-phosphate phosphatase to sphingosine [56]. It appears that an intricate balance should be maintained between products on each end of this pathway: ceramides and S1P [57,58]. Ceramides were implicated in neuronal death via the induction of caspase-9 and caspase-3 apoptosis [59,60]. Caspase independent mechanism of cell death, such as autophagy, have also been linked to increased ceramides [61,62]. Sphingosine, especially S1P, could regulate the egression of immune cells from lymphoid organs into circulation [63,64]. S1P binds to S1P receptor 1 (S1PR_1_), promoting T cells egression from the lymphoid tissue [65] and inflammation [66]. The role of S1P in pathogenesis of MS was also supported by the therapeutic efficacy of disease-modifying therapy targeting ligand S1PR_1_ [67].

We found evidence of disturbed glucose metabolism in MS. This is supported by the lower serum level of metabolites involved in gluconeogenesis and pyruvate pathways. Lower levels of metabolites of the glucose metabolism pathway could explain reports of lower glucose levels in MS compared to the control [68,69]. Interestingly, these changes varied between seasons. There were more glucose metabolic pathways affected in the fall. Metabolite analysis revealed evidence of the Warburg effect in fall and winter MS serum. Warburg effect refers to increased glycolysis and fermentation of pyruvate to lactate as an alternative to oxidation in the mitochondria [70]. This Warburg effect was shown to be utilized by activated T cells [71]. Additionally, glycolytic reprogramming was demonstrated to be a requirement for CD8+ T cells’ cytokine production and cytotoxic effect [72,73]. Glycolytic metabolism of activated CD8+ T cells was shown to correlate with lymphocyte proliferation [74]. Additionally, coordinated glycolytic reprogramming is required for CD8 effector functions such as cytokine transcription and cytotoxic activity [70,75]. Activated dendritic cells also switch to high rates of glycolysis [75,76]. It appears that pro-inflammatory Th1 and Th17 cell activity depends on glycolysis, while anti-inflammatory Tregs utilizes fatty acid oxidation for energy production [77]. Glycolysis is required for dendritic cell survival and migration [70,78]. The therapeutic potential of inhibiting glycolysis was tested in experimental autoimmune encephalitis models of MS. Inhibiting glycolysis was suggested as a mechanism for producing dimethyl fumarate therapeutic effects in MS [79]. 

Supporting previous findings, we have found affected amino acid pathways in MS [80,81]. Similar to Fitzgerald’s et al. data, we found disturbed aromatic amino acid metabolome [27]. Our data advance our understanding of seasonal changes in serum amino acids and their metabolites in MS. We found that a large number of these metabolites was lower in fall MS. In contrast, more amino acids and metabolites were higher in the summer and winter. We also demonstrated seasonal changes in serum level of L-Kynurenine and quinolinic acid, tryptophan metabolism products [82]. L-Kynurenine is an intermediate metabolite of tryptophan, which could be converted to kynurenic acid or quinolinic acid [82]. Quinolinic acid is identified as excitotoxic, damaging axons and dendrites and causing death of neurons [7,83,84,85,86]. High quinolinic/kynurenic acids ratio is demonstrated in relapsed RRMS, confirming a potential leading role of quinolinic acid in pathogenesis of MS [87]. We have found an increased quinolinic acid serum level in winter MS. This increased level of tryptophan metabolite could indicate activation of pathogenic metabolites, which could be an early warning sign of future relapse.

Our data support previous observations demonstrating low levels of vitD in MS [88]. Our data further advance our understanding of the role of vitD and its derivatives in MS pathogenesis. We have found that fall was the only season when the level of vitD was not lower than that in the control. In contrast, decreased vitD serum levels were demonstrated in the winter, spring, and summer. It was also an interesting observation that the level of quinolinic acid was affected in winter MS, which is also the season when we found a drop in vitD levels. In a recent study, the effect of vitD on levels of kynurenine pathway metabolites, quinolinic acid, and kynurenic acid, was demonstrated [89]. The authors showed that vitD supplements reduced production of both metabolites. In addition to a potential contribution to tryptophane metabolism, vitD supports the antioxidant pathways [90,91]. The antioxidant vitD may also be able to reduce excitotoxicity through its action on L-type calcium channels [92]. VitD could reduce excitotoxicity by reducing the number of calcium channels [93], reducing microglia activation [94], and increasing the production of anti-inflammatory cytokine IL-10 [95]. Our data suggest that decreased vitD levels and increased quinolinic acid levels during the winter could initiate development of the pro-inflammatory and neurotoxic environment, leading to exacerbation of MS symptoms.

Our data identified seasonal changes in metabolites levels in MS compared to the control. Many of these metabolites were previously found to be affected. Our study, for the first time, demonstrates seasonal changes in those metabolites in MS compared to the control. The largest number of metabolites affected was found to be in the fall followed by the spring. The smallest number of metabolite changes was in the summer. We found changes in lipid, carbohydrate, and amino acid metabolites in MS as compared to the control. Contribution of the affected metabolites is summarized in Figure 10. Ceramides, which are part of the sphingolipid pathway, were activated in all seasons, suggesting their central role in the disease pathogenesis. Substantial changes in glucose metabolite levels were found in MS, indicating a potential shift to glycolysis. An increased serum level of quinolinic acid was demonstrated in winter MS. Histidine pathways were affected, suggesting their role in relapse of MS in the spring and fall. We also found that spring and fall seasons had a higher number of overlapping metabolites affected in MS. This could be explained by patients having relapse of symptoms during these two seasons.

## 4. Materials and Methods

### 4.1. Study Subjects, Samples

Serum samples were collected from 48 RRMS cases (age 31.6 ± 8.3 years, 19 male and 29 female) admitted to the Republican Clinical Neurological Center (RCNC), in the Republic of Tatarstan, Russian Federation. Samples were collected from relapsed RRMS patients during their visit to the RCNC. MS reactivation was diagnosed during the spring and fall RCNC visits. Summer and winter MS samples were collected during remission. There were no cases of relapse diagnosed during the summer and winter. The average duration of the disease was between 3 to 5 years. RRMS was diagnosed based on clinical presentation and brain MRI results. Serum samples were collected from each patient and from 30 controls (age 35.2 ± 11.0 years, 15 male and 15 female). MS samples were collected throughout the year and grouped based on four seasons: fall (14 samples), winter (11 samples), spring (9 samples), and summer (14 samples). Control samples were collected in the fall (8 samples), winter (10 samples), spring (5 samples), and summer (7 samples). Each patient and control provided a single serum sample. Samples were collected before 1200 during their RCNC visit. Controls were self-identified as healthy individuals without any comorbidities, previous diagnosis of MS, or any other neurological diseases.

To collect the required serum samples, blood samples were collected into the Vacutest serum with gel (5 mL; Vacutest Kima, Arzergrande, Italy), mixed, and left to coagulate for 30 min. Serum was separated (1500 g for 10 min at 4 °C) and immediately used to make aliquots (100 µL). Serum samples were stored at −80 °C. Each serum sample was thawed once before analysis to avoid multiple freeze–thaw cycles. 

*Inclusion criteria*. RRMS diagnosis should be established by a neurologist according to the 2010 Revised MacDonald’s Diagnostic Criteria for MS [96]. MS patients were adults, male or female, without any comorbidities, including neurologic, cardiovascular, as well as metabolic diseases and cancer. The duration of the disease was no longer than 2 years.

For the control, inclusion criteria consisted of adult males and females without chronic diseases, including neurologic and metabolic diseases or cancer. Controls also required an absence of MS diagnosis and/or other autoimmune conditions within the family.

*Exclusion criteria*. Patients with neuromyelitis optica, other forms of MS, are receiving immunomodulatory therapy, or have an active infectious disease and treatment were excluded.

### 4.2. Metabolome Analysis

LS-MS/MS was used to complete targeted metabolomics analysis including 36 pathways, as described elsewhere [96,97,98], with modifications. In brief, serum from MS and controls was centrifuged (200× *g*; 10 min) and stored at −80 °C before use. To complete the analysis, methanol extracts of serum were collected and used in LC-MS/MS. LC-MS/MS was performed using high performance liquid chromatography (HPLC) Agilent 1260 Infinity II intracellular system combined with mass spectrometer tandem triple quadrupole mass spectrometer with linear ion trap (QTRAP 6500 plus (Sciex)) equipped with an electrospray ionization source ((ESI) Turbo V (Sciex)). Unlabeled standards and heavy standards were purchased from Sigma and Toronto Research Chemicals [99]. Unlabeled standards were used as quality control. The heavy standards were added to samples and analyzed together to make sure that retention time remains unaffected. The quality control (QC) samples contained a mixture of concentration-balanced standards according to plasma mean values. The QC samples were used as three technical replicates that were run three times before and after serum samples, and extracted water samples prepared the same way as serum samples served as blanks.

Metabolites were separated during chromatography with hydrophilic interaction (HILIC) using a polymer-based column with amino functional group Asahipak NH2P−40 2E, 4 μm, 250 × 2 mm (Shodex). The chromatographic separation conditions were as follows: mobile phase A: 95% H_2_O with 20 mM (NH_4_)_2_CO_3_ and 5% acetonitrile, pH 9.8; mobile phase B: 100% acetonitrile. The gradient included: 0–3.5 min 95% B, 3.6–8 min 85% B, 8.1–13 min 75% B, 14–30 min 0% B, 31–41 min 95% B, stop 46 min. The speed of flow was 200 μL/min with the injected volume of 10 μL.

Metabolites were measured in positive and negative ionization modes by rapid polarity reversal. MRM (multiple reactions monitoring) transitions selection (pairs of parent ions and their daughter fragments) and mass spectrometry parameter optimization were performed by direct infusion using solution of standards. The obtained ion chromatograms were viewed using the Analyst 1.6.1 software package (Sciex). Chromatographic peaks were verified by comparison with retention times and MRM-ratio of internal standards labeled with stable isotopes, as previously described [96]. Serum metabolites were measured by integration of chromatographic peaks area using Sciex OS v1.4.018067 (Sciex) and manually confirmed. There were 340 of 452 metabolites identified, and levels were measure in MS and control serum. 

### 4.3. Statistical Analysis

Metabolomic data were autoscaled (mean-centered and divided by the standard deviation of each variable) and the resulting scores analyzed by one-way ANOVA with pairwise comparisons and post hoc correction for multiple hypothesis testing using Fisher’s least significant difference method in MetaboAnalyst 5.0 (Available online: https://www.metaboanalyst.ca, accessed on 30 August 2022) [100]. SMPDB was selected as the metabolite set library for enrichment analysis.

## Figures and Tables

**Figure 1 ijms-24-03542-f001:**
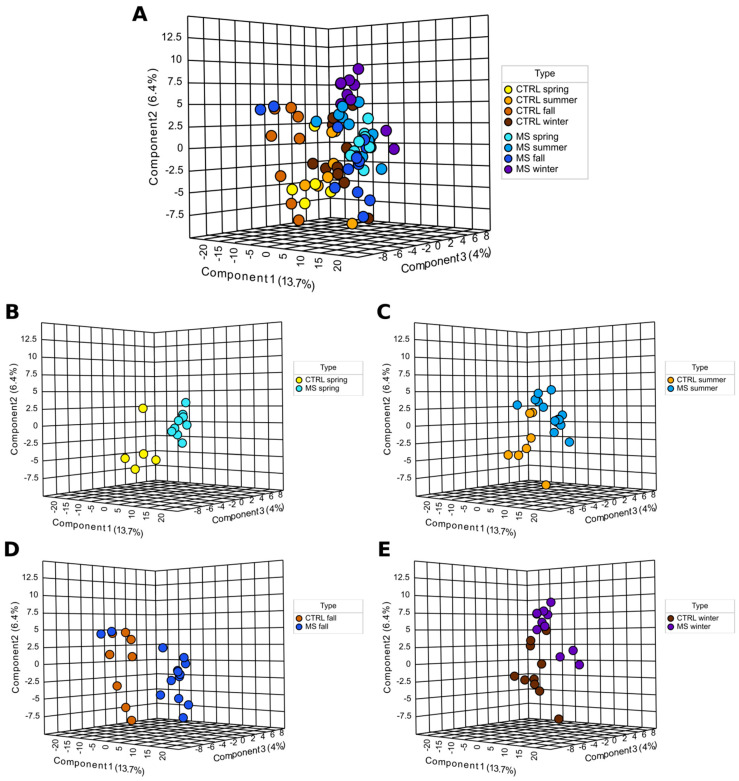
PLS-DA analysis of seasonal changes of metabolites in MS serum. PCA of seasonal level of metabolites in MS and control. This analysis is based on scaled levels of serum metabolites in MS and controls produced by MetaboAnalyst 5.0 (Available online: https://www.metaboanalyst.ca (accessed on 25 December 2022)). Each dot represents an MS patient or control. (**A**) Summary of all metabolites affected in all seasons. (**B**) Metabolites affected in Spring. (**C**) Metabolites affected in Summer. (**D**) Metabolites affected in Fall. (**E**) Metabolites affected in Winter.

**Figure 2 ijms-24-03542-f002:**
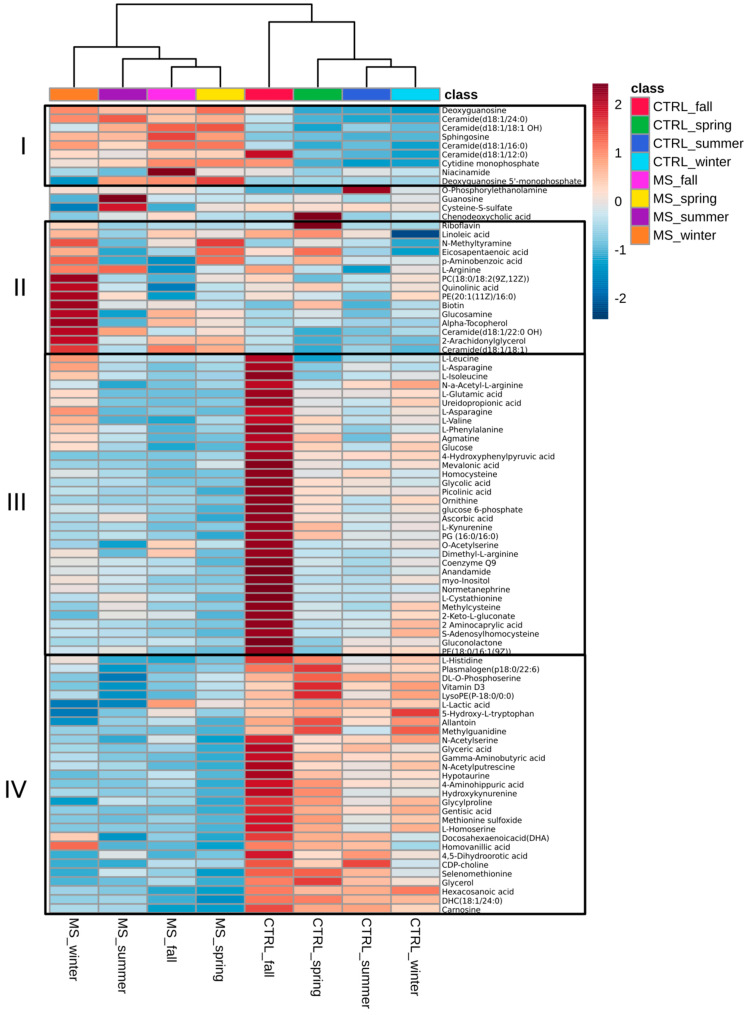
Heatmap analysis of metabolites in MS and control. Hierarchical cluster analysis was performed using Euclidean distance measure and Ward’s linkage clustering algorithm. Average values for each metabolite were used for analysis. Heatmap analysis is produced using MetaboAnalyst 5.0 (Available online: https://www.metaboanalyst.ca (accessed on 25 December 2022)). All presented metabolites differed significantly between at least one comparison group (p adj. < 0.05, one-way ANOVA with Fisher’s LSD).

**Figure 3 ijms-24-03542-f003:**
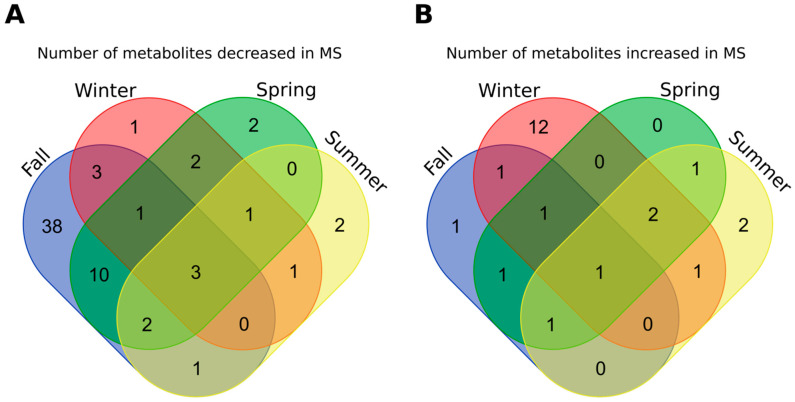
Venn diagram illustration of metabolites affected in MS. (**A**) Decreased metabolites in MS compared to control; (**B**) increased metabolites in MS compared to control.

**Figure 4 ijms-24-03542-f004:**
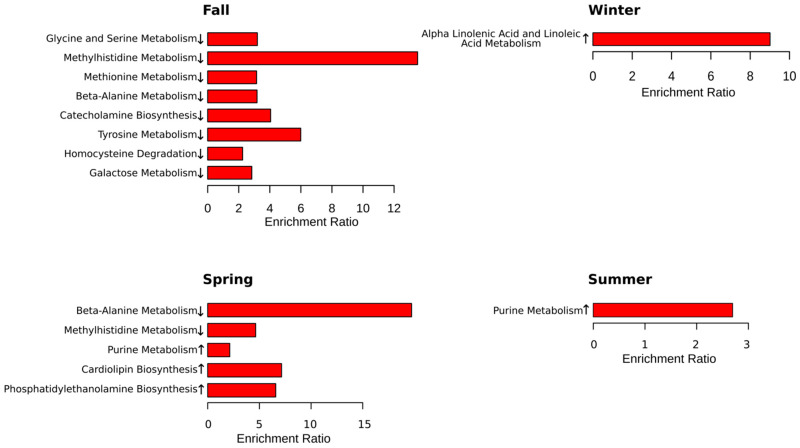
Metabolic pathways significantly enriched in each season. Enrichment analysis was performed in MetaboAnalyst 5.0 (Available online: https://www.metaboanalyst.ca (accessed on 8 November 2022)) using The Small Molecule Pathway Database (SMPDB, Available online: https://smpdb.ca (accessed on January 2014)). Enrichment ratio is computed by hits/expected, where hits = observed hits; expected = expected hits. ↑—increased serum level of metabolites included in the metabolic pathway in MS compared to control; ↓—decreased serum level of metabolites included in the metabolic pathway in MS compared to control.

**Figure 5 ijms-24-03542-f005:**
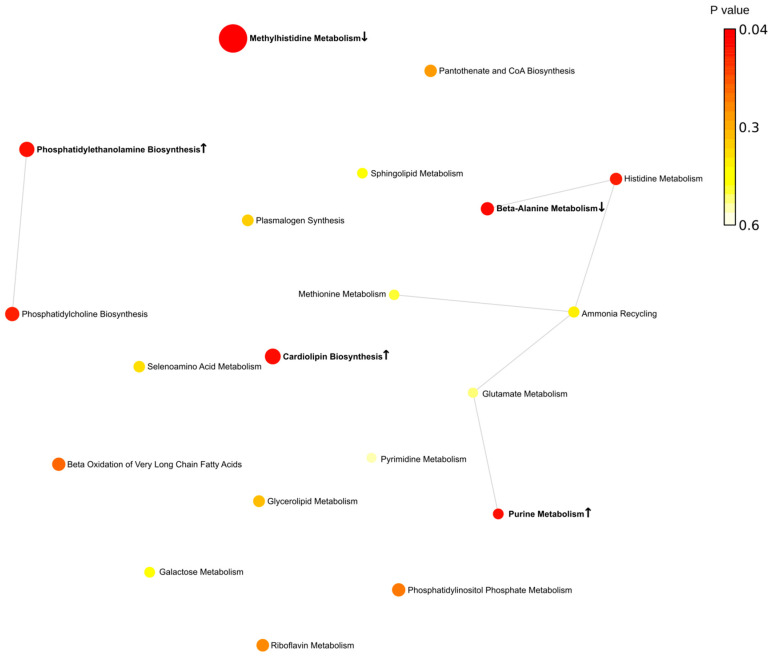
Metabolic pathways enriched in spring. Bold indicates significantly enriched pathways (*p* < 0.05). ↑—increased serum level of metabolites included in the metabolic pathway in MS compared to control; ↓—decreased serum level of metabolites included in the metabolic pathway in MS compared to control. Each node represents a metabolite set with its color based on its *p* value and its size based on fold enrichment. Two metabolite sets are connected by an edge if the number of their shared metabolites is over 25% of the total number of their combined metabolite sets. The yellow to red color gradient and larger size of the circle indicate lower *p* value.

**Figure 6 ijms-24-03542-f006:**
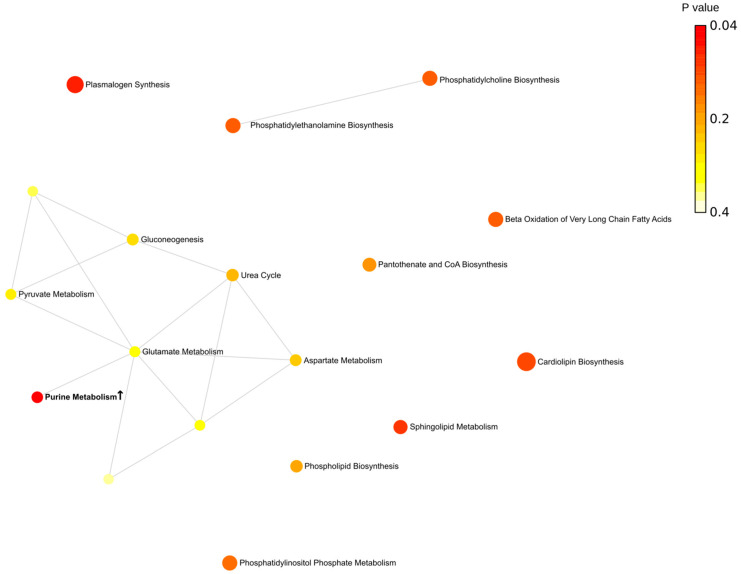
Metabolic pathways enriched in summer. Bold indicates significantly enriched pathways (*p* < 0.05). ↑—increased serum level of metabolites included in the metabolic pathway in MS compared to control. Each node represents a metabolite set with its color based on its *p* value and its size based on fold enrichment. Two metabolite sets are connected by an edge if the number of their shared metabolites is over 25% of the total number of their combined metabolite sets. The yellow to red color gradient and larger size of the circle indicate lower *p* value.

**Figure 7 ijms-24-03542-f007:**
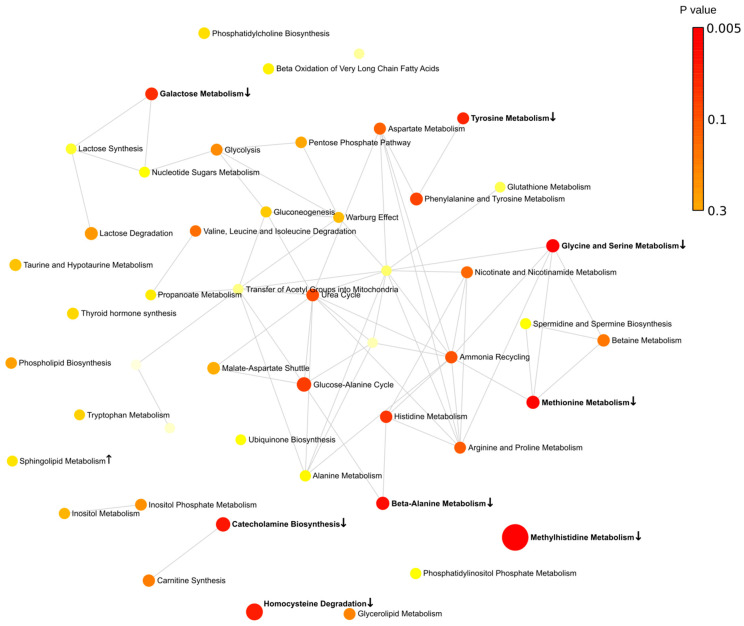
Metabolic pathways enriched in fall. Bold indicates significantly enriched pathways (*p* < 0.05). ↑—increased serum level of metabolites included in the metabolic pathway in MS compared to control; ↓—decreased serum level of metabolites included in the metabolic pathway in MS compared to control. Each node represents a metabolite set with its color based on its *p* value and its size based on fold enrichment. Two metabolite sets are connected by an edge if the number of their shared metabolites is over 25% of the total number of their combined metabolite sets. The yellow to red color gradient and larger size of the circle indicate lower *p* value.

**Figure 8 ijms-24-03542-f008:**
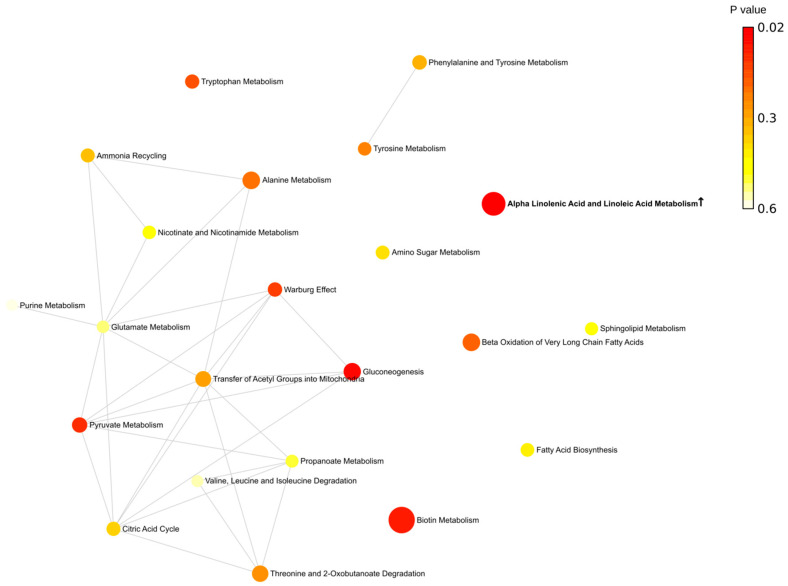
Metabolic pathways enriched in winter. Bold indicates significantly enriched pathways (*p* < 0.05). ↑—increased serum level of metabolites included in the metabolic pathway in MS compared to control. Each node represents a metabolite set with its color based on its *p* value and its size based on fold enrichment. Two metabolite sets are connected by an edge if the number of their shared metabolites is over 25% of the total number of their combined metabolite sets. The yellow to red color gradient and larger size of the circle indicate lower *p* value.

**Figure 9 ijms-24-03542-f009:**
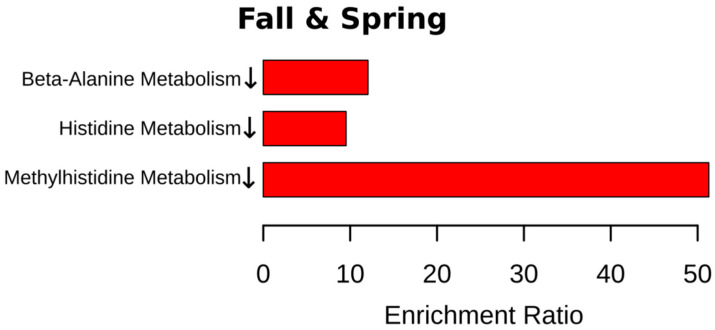
Metabolic pathways including metabolites similarly affected in spring and fall MS. Enrichment analysis was perform in MetaboAnalyst 5.0 (Available online: https://www.metaboanalyst.ca (accessed on 8 November 2022)) using The Small Molecule Pathway Database (SMPDB, Available online: https://smpdb.ca (accessed on January 2014)). Enrichment ratio is computed by hits/expected, where hits = observed hits; expected = expected hits.

**Figure 10 ijms-24-03542-f010:**
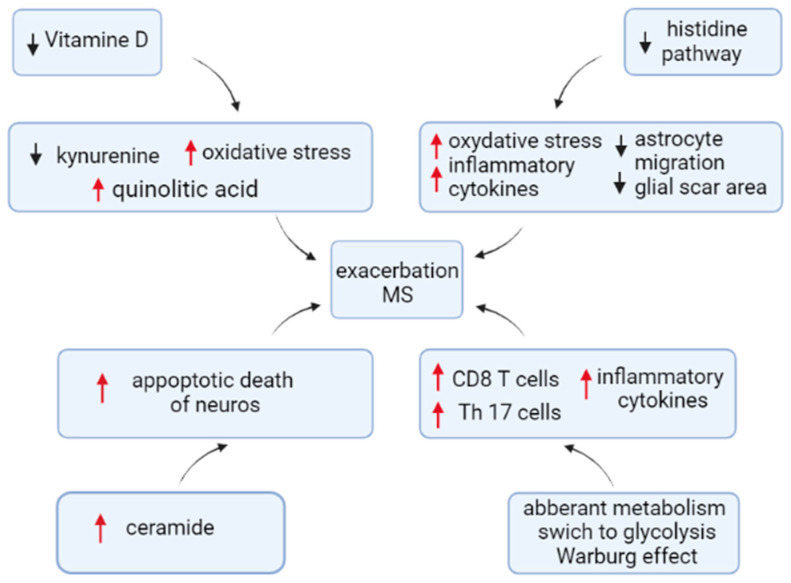
Contribution of metabolites to MS pathogenesis. Our data support the hypothesis of multifactorial pathogenesis of MS. Multiple metabolites were found affected in MS. These metabolites could contribute to oxidative stress, apoptotic death of neurons, astrocyte migration, and scar formation. In addition, affected metabolites could contribute to the activation of pro-inflammatory cytokine production and differentiation of pathogenic CD8+ and Th17 lymphocytes. Additionally, affected metabolites indicate changes in cell metabolism, which could support the differentiation of CD8+ cells.

**Table 1 ijms-24-03542-t001:** List of metabolites significantly changed in MS group in comparison to controls.

Decreased in MS Compared to CTRL			Increased in MS Compared to CTRL		
Seasons	Total Number of Shared Metabolites	Metabolite Names	Seasons	Total Number of Shared Metabolites	Metabolite Names
Fall Spring Summer Winter	3	Hexacosanoic acid	Fall Spring Summer Winter	1	Sphingosine
		DHC (18:1/24:0)	Fall Spring Winter	1	Ceramide (d18:1/16:0)
		Gentisic acid	Fall Spring Summer	1	Ceramide (d18:1/18:1 OH)
Fall Spring Winter	1	Glycylproline	Spring Summer Winter	2	Ceramide (d18:1/24:0)
Fall Spring Summer	2	Gamma-Aminobutyric acid			Deoxyguanosine
		Docosahexaenoic acid (DHA)	Fall Winter	1	Ceramide (d18:1/18:1)
Spring Summer Winter	1	Vitamin D3	Fall Spring	1	Deoxyguanosine 5’-monophosphate
Fall Winter	3	4-Hydroxyphenylpyruvic acid	Summer Winter	1	Ceramide (d18:1/22:0 OH)
		4,5-Dihydroorotic acid	Spring Summer	1	Cytidine monophosphate
		N-Acetylserine	Fall	1	Niacinamide
Fall Spring	10	Selenomethionine	Winter	12	Linoleic acid
		L-Histidine			Alpha-Tocopherol
		Carnosine			Cer(d18:1/12:0)
		4-Aminohippuric acid			N-Methyltyramine
		Agmatine			Eicosapentaenoic acid
		Hydroxykynurenine			Homovanillic acid
		LysoPE (P-18:0/0:0)			Biotin
		Methionine sulfoxide			2-Arachidonylglycerol
		Glycerol			Glucosamine
		Plasmalogen (p18:0/22:6)			PE (20:1(11Z)/16:0)
Fall Summer	1	CDP-choline			Quinolinic acid
Spring Winter	2	Allantoin			PC (18:0/18:2(9Z,12Z))
		Methylguanidine	Summer	2	L-Arginine
Summer Winter	1	L-Lactic acid			Guanosine
Fall	38	Glucose 6-phosphate			
		N-Acetylputrescine			
		Gluconolactone			
		Cer (d18:1/12:0)			
		L-Isoleucine			
		Picolinic acid			
		N-a-Acetyl-L-arginine			
		Anandamide			
		L-Asparagine			
		L-Valine			
		Ornithine			
		L-Arginine			
		Glyceric acid			
		Glucose			
		2-Keto-L-gluconate			
		Myo-Inositol			
		Homovanillic acid			
		L-Cystathionine			
		Glycolic acid			
		S-Adenosylhomocysteine			
		PE (18:0/16:1(9Z))			
		2 Aminocaprylic acid			
		Mevalonic acid			
		Normetanephrine			
		Homocysteine			
		Methylcysteine			
		Coenzyme Q9			
		L-Homoserine			
		O-Acetylserine			
		Ascorbic acid			
		L-Leucine			
		Ureidopropionic acid			
		PG (16:0/16:0)			
		Dimethyl-L-arginine			
		L-Glutamic acid			
		Hypotaurine			
		L-Kynurenine			
		L-Phenylalanine			
Winter	1	5-Hydroxy-L-tryptophan			
Spring	2	Riboflavin			
		Chenodeoxycholic acid			
Summer	2	O-Phosphorylethanolamine			
		DL-O-Phosphoserine			

## Data Availability

The data presented in this study are available on request from the corresponding author.

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
