# Peer review of "Seasonal Changes in Serum Metabolites in Multiple Sclerosis Relapse"

_ijms, 2023, doi:10.3390/ijms24043542_

Round 1
Reviewer 1 Report
This is a timely and interesting study. Authors should be aware of and should cite following relevant clinical trials: NCT03266965, NCT04764383. The authors should also incorporate selected references that form the background of the above clinical trials. Authors should consider the following paper: Nat Metab. 2022 Sep;4(9):1086-1088. doi: 10.1038/s42255-022-00628-3. PMID: 35934691 to rewrite or modify their methods section. Authors should clearly delineate the composition and concentration of the internal standards used for identification. Authors could consider use of other software such as MS-DIAL 4.9, CompoundDiscoverer3.2, Skyline or other software to at their discretion to ensure if usage of additional software confirm/expand their findings. Overall, their manuscript needs light English editing, brevity and improvment in presentation. Authors, at their discretion may also consider incorporating a summary cartoon as part of Figure 10 or alternately may consider making the Figure 10 as a more appealing summary figure for the readership.
Author Response
This is a timely and interesting study. Authors should be aware of and should cite following relevant clinical trials: NCT03266965, NCT04764383. The authors should also incorporate selected references that form the background of the above clinical trials.
Agree: clinical trials are cited: lines 291-292
Authors should consider the following paper: Nat Metab. 2022 Sep;4(9):1086-1088. doi: 10.1038/s42255-022-00628-3. PMID: 35934691 to rewrite or modify their methods section.
Answer: in manuscript by McDonnald et al (doi: 10.1038/s42255-022-00628-3.), authors conducted the widescreen lipidomics analysis. This analysis includes lipids only. In contrast, our study included variety of molecules, not only lipids. These molecules were amino acids, proteins, organic acids, lipids, etc. Therefore, we used Targeted Mass Spectrometry (MRM/SRM) described by Li, K., et al., (A robust, single-injection method for targeted, broad-spectrum plasma metabolomics. Metabolomics, 2017. 13(10): p. 122.)
Authors should clearly delineate the composition and concentration of the internal standards used for identification.
Answer: In our study, we did conducted quantification of metabolites. Instead, “serum metabolites were measured by integration of chromatographic peaks area using Sciex OS v1.4.018067 (Sciex) and it was manually confirmed”. Heavy standards were used to demonstrated the quality of run, not for quantitation of each metabolite. We followed the protocol described by Li, K., et al., (A robust, single-injection method for targeted, broad-spectrum plasma metabolomics. Metabolomics, 2017. 13(10): p. 122.).
To clarify the metabolite quantification, changes were made in the
Authors could consider use of other software such as MS-DIAL 4.9, CompoundDiscoverer3.2, Skyline or other software to at their discretion to ensure if usage of additional software confirm/expand their findings.
Answer: In our study, we used targeted metabolome analysis described by Naviaux, R.K., et al., (Metabolic features of Gulf War illness. PloS one, 2019. 14(7): p. e0219531;. Li, K., et al., Improved Dried Blood Spot-Based Metabolomics: A Targeted, Broad-Spectrum, Single-Injection Method. Metabolites, 2020. 10(3).). The program used for this analysis is recommended by the Sciex equipment supplier, who provided the SiexOS and MultiQuant Software programs.
The MS -DIAL 4.9 software is not optimized for targeted metabolomics; instead it states that this program is designed for untargeted metabolome analysis (https://mtbinfo-team.github.io/mtbinfo.github.io/MS-DIAL/tutorial.html). Therefore, we followed the manufacturer’s recommendation and used Sciex OS v1.4.018067 (Sciex) software.
Overall, their manuscript needs light English editing, brevity and improvment in presentation.
Agree: substantial English editing was completed
Authors, at their discretion may also consider incorporating a summary cartoon as part of Figure 10 or alternately may consider making the Figure 10 as a more appealing summary figure for the readership.
Agree: a summary figure was added as figure 10. We moved the old Figure 10 to supplementary material section. It is now labeled as Supplementary figure 1.
Reviewer 2 Report
This study by Martynova et al. examines seasonal changes in a vast array of metabolites from serum samples of MS patients compared to controls, in relation to known seasonal exacerbation of clinical symptoms. Using liquid chromatography, they found more metabolites were affected (either up- or downregulated) in MS patients in the fall and spring, while summer was characterized by a small number of changes. The novelty of this study resides in the examination of changes in metabolites by season. I find the figures and tables highly informative and elegant, and the text is easy to follow. English language is solid, except for a couple of misspelling easily correctable.
I have 2 main comments on the study:
11) I am not sure how strong the sample size is to derive strong conclusions. It seems they only collected 1 sample per subject in 48 MS patients and 30 controls. And these are further divided in the different seasons: fall (n=14 MS, 8 C-); winter (n=11 MS, 10 C-); spring (n=9 MS, 5 C-); and summer (n=14 MS, 7 C-). Again, I leave the decision to the Editors to ponder if this constitutes sufficient data for the Journal standards. For example, what is the sample size in similar studies? In the Salvi et al. study they recruited 96 patients; however, in the Harding study they recruited 2076.
22) This study is quite observational rather than mechanistical. For this reason, I suggest the authors tone down a bit the nature of their conclusions. For example, the title states “Seasonal changes in serum metabolites indicate histidine pathways as playing role in Multiple Sclerosis relapse”. This sounds quite categorical, nonetheless I do not see how such inference can be elucidated from the type of study the authors carried out (descriptive, observational). I suggest they rework the title and main conclusions to allow room for alternative interpretations.
Author Response
This study by Martynova et al. examines seasonal changes in a vast array of metabolites from serum samples of MS patients compared to controls, in relation to known seasonal exacerbation of clinical symptoms. Using liquid chromatography, they found more metabolites were affected (either up- or downregulated) in MS patients in the fall and spring, while summer was characterized by a small number of changes. The novelty of this study resides in the examination of changes in metabolites by season. I find the figures and tables highly informative and elegant, and the text is easy to follow. English language is solid, except for a couple of misspelling easily correctable.
I have 2 main comments on the study:
11) I am not sure how strong the sample size is to derive strong conclusions. It seems they only collected 1 sample per subject in 48 MS patients and 30 controls.
Answer: MS is rare disease in Republic of Tatarstan. Therefore, there is a limitations in how many samples will be available for study. The number of samples used in our study is within the range used in similar studies of metabolome in MS. For example, Villoslada et al used serum samples from 61 MS patients for metabolome analysis (DOI: https://doi.org/10.1212/NXI.0000000000000321). In another study by Zahor et all published in PNAS, 35 RRMS patients were recruited for study metabolomic (https://doi.org/10.1073/pnas.212326511). One more study by Yeo et all, published in Scientific Reports, serum samples from 31 RRMS and 28 SPMS patients were used for study the metabolome (https://www.nature.com/articles/s41598-020-69119-3).
And these are further divided in the different seasons: fall (n=14 MS, 8 C-); winter (n=11 MS, 10 C-); spring (n=9 MS, 5 C-); and summer (n=14 MS, 7 C-). Again, I leave the decision to the Editors to ponder if this constitutes sufficient data for the Journal standards. For example, what is the sample size in similar studies?
Answer: In a similar study by Villoslada et al, serum samples from 61 MS patients were used for metabolome analysis (DOI: https://doi.org/10.1212/NXI.0000000000000321). In another study by Zahor et all published in PNAS, 35 RRMS patients were recruited for study metabolomic (https://doi.org/10.1073/pnas.212326511). One more study by Yeo et all, published in Scientific Reports, serum samples from 31 RRMS and 28 SPMS patients were used for study the metabolome (https://www.nature.com/articles/s41598-020-69119-3). Therefore, we believe that number of samples used in our study is within the range used by researchers.
In the Salvi et al. study they recruited 96 patients; however, in the Harding study they recruited 2076.
It should be noted that there are limited number of MS patients in the Republic of Tatarstan. The prevalence and incidence rate of MS is 36.7 and 5.5, respectively (https://www.ncbi.nlm.nih.gov/pmc/articles/PMC7288193/). These values are within the medium zone of MS prevalence according to Wade, BJ (https://www.hindawi.com/journals/msi/2014/124578/). That is in the striking contrast with the MS prevalence and incidence in England, where it could be as high as 203.4 and 9.64 respectively (https://jnnp.bmj.com/content/85/1/76). This puts England into the very high MS prevalence zone, according to Wade, BJ classification. Therefore, it is more feasible to recruit large number of patients in England as it was done by Harding et al (https://pubmed.ncbi.nlm.nih.gov/28424900/). Also, in Harding et al study, samples were collected during the period between 2005 and 2014. We have collected samples only during a single year to make sure that our data would not be affected by yearly changes in weather (i.e. El Ninio, etc).
MS prevalence in Italy is 113-141 cases per 100,000 population (https://doi.org/10.2147/CLEP.S376005). This is the high risk of MS zone (https://www.hindawi.com/journals/msi/2014/124578/). Having prevalence in Italy higher than in Tatarstan (113 vs 36.7, respectively), Salvi et al collected 96 samples, while we have included 48 samples.
We also would like to point is that neither Harding et al nor Salvi et al included metabolome data. All data used was seasonal variation s in relapse, monthly hours of sunshine, demographic data, temperature, humidity and wind speed.
22) This study is quite observational rather than mechanistical. For this reason, I suggest the authors tone down a bit the nature of their conclusions. For example, the title states “Seasonal changes in serum metabolites indicate histidine pathways as playing role in Multiple Sclerosis relapse”. This sounds quite categorical, nonetheless I do not see how such inference can be elucidated from the type of study the authors carried out (descriptive, observational). I suggest they rework the title and main conclusions to allow room for alternative interpretations.
Agree: We changed the title of this manuscript simplifying it to: Seasonal changes in serum metabolites in Multiple Sclerosis relapse
Reviewer 3 Report
Comments to the authors
This study examined that distribution of seasonal changes in serum metabolites indicate histidine pathway as playing role in multiple sclerosis relapse. The purpose of this manuscript was to analyze the seasonal variation of serum metabolite levels in MS compared to the control. However, the authors should employ two-way ANOVA (differences among treatments were analyzed as a 2×4 (MS×Season) factorial arrangement to discuss relationship between seasonal changes in metabolites and MS morbidity. In addition, the authors claimed that metabolites of histidine pathway during the spring and fall, which could contribute to MS exacerbations. However, there is no information why histidine metabolites changed significantly either in spring or fall. The reviewer considers that the authors need to discuss this reason. Overall, the reviewers consider that this paper does not reach the level of a scientific publication.
Author Response
This study examined that distribution of seasonal changes in serum metabolites indicate histidine pathway as playing role in multiple sclerosis relapse. The purpose of this manuscript was to analyze the seasonal variation of serum metabolite levels in MS compared to the control. However, the authors should employ two-way ANOVA (differences among treatments were analyzed as a 2×4 (MS×Season) factorial arrangement to discuss relationship between seasonal changes in metabolites and MS morbidity.
Answer: We used the one-way ANOVA because it allows identifying the differences between the large number of studied comparison groups (8 groups in our study). The comparison groups should have the same sample size to be suitable for the two-way ANOVA analysis. In our study, we had different number of samples in each group. Therefore, we used one-way ANOVA with pairwise comparisons and posthoc correction for multiple hypothesis testing.
Using one-way ANOVA analysis we were able to demonstrate differences in metabolites between MS and control in each season. We were also able to demonstrate differences in metabolite serum level between MS and control independent of season.
In addition, the authors claimed that metabolites of histidine pathway during the spring and fall, which could contribute to MS exacerbations. However, there is no information why histidine metabolites changed significantly either in spring or fall. The reviewer considers that the authors need to discuss this reason. Overall, the reviewers consider that this paper does not reach the level of a scientific publication.
Agree: we added the plausible explanation on why histidine metabolites could be changed during fall and spring; lines 293-299
“Our understanding of the seasonal variations of histamine serum level in MS is limited. Studies have identified histidine derivative, trans-urocanic acid, as a photoreceptor in stratum corneum of skin [44-45]. Ultraviolet light exposure produces cis-urocanic acid, a soluble molecule which could stimulate oxidative radicals production and DNA damage [46-47]. It should be noted that histidine levels change in MS during fall and spring, seasons when sunlight exposure changes substantially. However, the precise mechanisms of these changes in histidine level in MS remain unknown.”
Reviewer 4 Report
The study conducted titled “Seasonal changes in serum metabolites indicate histidine pathway as playing role in Multiple Sclerosis relapse” is interesting and is directed toward the season effects on the relapse of Multiple Sclerosis where the study further probes into the mechanisms behind. Cytokines expression is being studied with its correlation to season previously. The study identified the “Spring & Fall” seasons where relapse is reported and higher levels of overlapping metabolites in these two seasons.
The following suggestions need consideration and to be aligned in the revised version of the Manuscript
· The last paragraph of the Introduction part gives the sense of concluding remarks, where the Aim of the study need to be established at the end of Introduction section. The authors are suggested to revise the last paragraph (Line 75 – 83) and establish the Aim of the study, while the description of study results must be avoided.
· Sample size? How is the sample size calculated and how the authors concluded that the sample size in the study is having the capability to detect the significant differences?
· Male vs female ratio? The Male Vs Female ratio is having important role in the interpretation of study outcomes and the results. How the current study considers the Male and Female ratios; Are these dependent variables or independent variables?
· Study duration? What will be the impact if the study duration is changed? How the outcomes of the study will be affected? Is the study duration in the current study is fine?
Author Response
The study conducted titled “Seasonal changes in serum metabolites indicate histidine pathway as playing role in Multiple Sclerosis relapse” is interesting and is directed toward the season effects on the relapse of Multiple Sclerosis where the study further probes into the mechanisms behind. Cytokines expression is being studied with its correlation to season previously. The study identified the “Spring & Fall” seasons where relapse is reported and higher levels of overlapping metabolites in these two seasons.
The following suggestions need consideration and to be aligned in the revised version of the Manuscript
- The last paragraph of the Introduction part gives the sense of concluding remarks, where the Aim of the study need to be established at the end of Introduction section. The authors are suggested to revise the last paragraph (Line 75 – 83) and establish the Aim of the study, while the description of study results must be avoided.
Agree: changes were made in the text lines80-84
- Sample size? How is the sample size calculated and how the authors concluded that the sample size in the study is having the capability to detect the significant differences?
Answer: MS is rare disease in Republic of Tatarstan. Therefore, there is a limitations in how many samples will be available for study. The number of samples used in our study is within the range used in similar studies of metabolome in MS. For example, Villoslada et al used serum samples from 61 MS patients for metabolome analysis (DOI: https://doi.org/10.1212/NXI.0000000000000321). In another study by Zahor et all published in PNAS, 35 RRMS patients were recruited for study metabolomic (https://doi.org/10.1073/pnas.212326511). One more study by Yeo et all, published in Scientific Reports, serum samples from 31 RRMS and 28 SPMS patients were used for study the metabolome (https://www.nature.com/articles/s41598-020-69119-3).
- Male vs female ratio? The Male Vs Female ratio is having important role in the interpretation of study outcomes and the results. How the current study considers the Male and Female ratios; Are these dependent variables or independent variables?
Answer: The male to female ratio of MS patients was 1:1.5, supporting the previous observation that this disease is more often diagnosed in female compared to male. In controls the ratio was 1:1. The ratio male/female in MS and controls did not differ significantly. These variables are independent.
- Study duration? What will be the impact if the study duration is changed? How the outcomes of the study will be affected? Is the study duration in the current study is fine?
Answer: we planned to conduct this study for one year to exclude the potential effect of climate differences between years (duration of the sun light, humidity, number of rainy days, seasonal changes in temperature, El Ninio effect, etc). There is a possibility that the level of metabolites could vary from year to year; however, we believe that overall, the trend of metabolite levels in controls and MS will remain similar to what we found in this study.
In most of the MS metabolome studies MS serum samples were collected once during various periods of time. In study by Zahoor et al, samples were collected during a single year (https://www.pnas.org/doi/10.1073/pnas.2123265119). In another study by Lim et al, samples were collected during 2 years (https://www.nature.com/articles/srep41473). Nourbakhsh et al were collecting samples for 5 years (https://onlinelibrary.wiley.com/doi/full/10.1002/acn3.637).
Round 2
Reviewer 3 Report
Comments to the author
The authors completely revised their manuscript. Therefore, I recommend it for acceptance.